# Assessment of peak bone mineral density and its associated factors in Vietnamese adults: A cross-sectional study

Hoang Thanh Van Nguyen[1]*, Thi Thu Ha Tran[1], Huu Quyet Le[2], Hoang Minh Nguyen[3], Thi Hong Van Le[1]

1 Department of Internal Medicine, University of Medicine and Pharmacy, Hue University, Hue City, Vietnam, 2 Department of Anatomy – Experimental Surgery, University of Medicine and Pharmacy, Hue University, Hue City, Vietnam, 3 Department of Diagnostic Imaging, Hue Medic Medical Center, Hue City, Vietnam

* hntvan@hueuni.edu.vn, nhtvan@huemed-univ.edu.vn

## Abstract

### Introduction

Osteoporosis is a growing public health concern in Vietnam, yet population-specific reference data for peak bone mineral density (PBD) remain limited. This study aimed to establish a standard PBD dataset and identify factors associated with bone mineral density (BMD) in Vietnamese adults.

### Methods

A cross-sectional study included 1,378 participants (410 men, 968 women) in Hue City, Vietnam. BMD was measured at the lumbar spine (LS), femoral neck (FN), and total hip (TH) using dual-energy X-ray absorptiometry (DXA). A cubic polynomial regression models were performed to identify peak bone density (PBD). Age-standardized prevalence of osteoporosis was calculated using the 2024 Vietnamese population structure.

### Results

Men exhibited higher BMD than women across all skeletal sites. The estimated age of PBD attainment was 20–29 years in men and approximately 30 years in women. Age was the strongest negative predictor of BMD, while body weight and height showed positive correlations. The age-standardized prevalence of osteoporosis was highest at the LS (33.58%), followed by the TH (12.77%) and FN (2.69%). Women showed a markedly higher prevalence than men, with a sharp increase observed after menopause.

**Data availability statement:** All revelant data are within the manuscript and its Supporting Information files.

**Funding:** The author(s) received no specific funding for this work.

**Competing interests:** The authors have declared that ni competing interests exist.

## Conclusion

This study provides an updated reference dataset for PBD in the Vietnamese population, notably revealing that men attain peak bone density earlier than women. Furthermore, the findings underscore the high prevalence of osteoporosis at the lumbar spine, suggesting a need for early screening strategies targeting high-risk groups.

## Introduction

Osteoporosis is a systemic bone disease characterized by decreased bone mass and microarchitectural deterioration of bone tissue, leading to increased bone fragility and fracture risk. Consequently, it is now recognized not only as an individual medical problem but as a global public health challenge [1,2]. Bone density is considered a fundamental determinant of skeletal strength and directly influences the risk of osteoporosis and fractures in middle-aged and older adults. However, the WHO T-score relies on the Peak Bone Mineral Density (PBMD) of young White women (US NHANES III) [3,4]. Applying this biological reference to Asians, and specifically the Vietnamese, is problematic due to differences in skeletal size and genetics. Despite having lower BMD than Whites, Asians have lower rates of hip fracture a discrepancy termed the 'Asian Paradox [5]. Consequently, using White reference populations may misrepresent the actual bone quality of Vietnamese people, resulting in overdiagnosis and unwarranted treatment [6].

A foundational study by Nguyen et al. (2009) defined BMD reference ranges and an age at peak BMD of 27–29 years, however, this study focused only on women [7]. This omission is significant because men suffer a mortality rate after hip fracture that is double that of women [8]. According to Huy G. Nguyen and colleagues, PBD is higher in men than in women, with the difference being more pronounced at the femoral neck (mean difference: 0.123 g/cm²) than at the lumbar spine (mean difference: 0.019 g/cm²). The study also demonstrated a strong association between lean mass and BMD [9]. In particular, sex-specific differences in bone mass accumulation and the influence of body composition (muscle and fat mass) on PBMD have not been fully elucidated in recent reports [9].

In Vietnam, osteoporosis is emerging as a critical public health challenge due to an unprecedented demographic transition. This shift is marked by rapid population aging from 2015 to 2035, with the country projected to become a fully aged society after 2036 [10]. This demographic shift is driving a rise in non-communicable diseases, particularly musculoskeletal disorders which threaten the quality of life of the elderly and place an immense strain on social security and healthcare systems [8]. Regarding osteoporosis specifically, while available data indicate a prevalence of 27% in women and 13% in men, research in Vietnam has predominantly focused on the female population [11].

Multiple factors-including sex, genetics, lifestyle, diet, and physical activity-contribute to bone mass accrual and loss [12–14]. Age, body weight, and height are

all significant predictors of BMD; however, the study by Lin YC et al (2003) demonstrated that body weight is a stronger predictor than age across all measured sites [15].

To address these gaps, this study aims to establish a standard PBMD dataset specific to the Vietnamese population, identify factors associated with BMD, and provide an accurate assessment of the osteoporosis burden by sex. These findings are intended to serve as a foundation for adjusting clinical diagnostic and treatment guidelines and provides essential direction for early osteoporosis prevention strategies.

## Subject and method

### Ethical approval

The study was approved by the Institutional Ethics Committee of Hue University of Medicine and Pharmacy (approval code: H2025/453). All participants were fully informed about the study's objectives and procedures and provided written informed consent prior to enrollment. Personal information was kept strictly confidential and used solely for scientific research purposes.

### Study design

A cross-sectional survey.

### Study population

A total of 1,378 healthy adult men and women who met the study's inclusion and exclusion criteria were recruited and categorized into seven age groups (<20; 20–29; 30–39; 40–49; 50–59; 60–65; >65 years) and by sex (men: n = 410; women: n = 968).

All participants underwent BMD measurements at Hue University of Medicine and Pharmacy Hospital between June 2025 and September 2025.

**Inclusion and exclusion criteria.** Inclusion criteria encompassed adults aged 18 years and older who provided consent to participate. To establish a healthy reference population, strict exclusion criteria were applied. Participants were excluded if they had a history of fractures, prior osteoporosis treatment, or any medical conditions affecting bone metabolism (e.g., hyperthyroidism, hyperparathyroidism, chronic renal failure, rheumatoid arthritis, or malabsorption syndromes). Additionally, individuals using medications known to influence bone density; those who had been bedridden for more than one month, and pregnant women were excluded from the study.

### Sample size and sampling

The sample size was calculated using the formula for estimating a population mean, with a 95% confidence level and a permissible error of 5%. Based on preliminary data and prior studies on the variability of bone mineral density (BMD) at DXA measurement sites (lumbar spine and femoral neck), a minimum of 100 participants for each major group (by sex) was determined to be necessary. To enhance reliability and compensate for potential attrition, a total of 1,378 participants were enrolled, including 410 men and 968 women, distributed across seven age groups. Participants were selected using a convenience sampling method from healthy adults undergoing routine health examinations or BMD measurements at Hue University of Medicine and Pharmacy Hospital, provided they met the study's inclusion and exclusion criteria.

Data collection occurred from 10 June 202530 September 2025.

### Variables and measurements

Anthropometric measurements were taken with participants wearing light indoor clothing and no shoes. Standing height was measured to the nearest 0.1 cm using a wall-mounted stadiometer, and weight was measured to the nearest 0.1 kg

using an electronic scale. Body Mass Index (BMI) was calculated as weight in kilograms divided by the square of height in meters ($kg/m^2$).

Bone Mineral Density (BMD; $g/cm^2$) was measured at the lumbar spine (L1–L4), femoral neck, and total hip using Dual-energy X-ray Absorptiometry (DXA) (Horizon Wi, Hologic, USA). The densitometer was calibrated daily using a standard phantom. The precision error, expressed as the coefficient of variation (CV), was 1.5% for the lumbar spine and 1.2% for the hips.

Additionally, osteoporosis status was defined as a binary outcome variable. Participants were classified as having osteoporosis if their T-score at the femoral neck was ≤ −2.5. T-scores were calculated based on the peak bone mass of the young adult reference group (aged 20–29 years) derived specifically from this study population.

## Statistical analysis

Baseline characteristics of the study population were summarized using descriptive statistics; continuous variables were expressed as means and standard deviations (SD), while categorical variables were presented as frequencies and percentages. Comparisons between groups were performed using ANOVA. Pearson's correlation coefficients were calculated to assess the linear relationships between BMD at the lumbar spine, femoral neck, and total hip with age and anthropometric indices. A p-value of less than 0.05 was considered statistically significant.

To estimate peak bone mass and the age at which it is attained, BMD data were fitted to a cubic polynomial regression model as a function of age:

$$BMD = \alpha + \beta_1 (Age^1) + \beta_1 (Age^2) + \beta_1 (Age^3)$$

The age of peak BMD was determined by solving the first derivative of the regression equation for zero. The 95% confidence interval (CI) for the age of peak BMD was estimated using the bootstrap method with 1,000 replications.

For the male cohort, as the regression models did not yield a discernible peak within the study age range, the mean and SD of the young adult group (aged 20–29 years) were utilized as reference values. T-scores were calculated for each individual based on these internal reference data using the formula:

$$T-scores = \frac{Individual\ BMD\ -\ Mean\ Peak\ BMD)}{SD\ of\ Peak\ BMD.}$$

Osteoporosis was defined according to the World Health Organization criteria (*T-score* ≤ −2.5 at the femoral neck). Finally, the prevalence of osteoporosis was age-standardized to the Vietnamese population structure (year of 2024) using the direct standardization method.

All data analysis were performed using Stata version 17 (StataCorp, TX, USA).

## Results

Table 1 show the information on anthropometric characteristics and bone mineral density (BMD) values by assessing across these groups, with additional analyses performed separately for men and women (S1 Table 1 and S2 Table).

In the overall study population, mean body weight declined from 56.5 kg in individuals younger than 20 years to 50.8 kg in those aged ≥65 years. Height showed a similar downward trajectory, decreasing from 164.0 cm to 151.1 cm across the same age range. BMI increased modestly from 20.91 in the youngest group to a peak of 23.13 in the 50–59 age group, before slightly declining in older participants (Table 1).

BMD values at the LS, TH, and FN declined progressively with age in the overall population (Table 1). At the LS, mean BMD decreased from 0.981 g/cm² in individuals <20 years to 0.742 g/cm² in those ≥65 years. At the TH, values fell from 1.024 g/cm² to 0.777 g/cm², while at the FN, BMD declined from 0.977 g/cm² to 0.662 g/cm². All differences across age groups were statistically significant (p < 0.001) (Table 1).

                                                            

**Table 1. Characteristics of study subjects.**

| Age group | <20 | 20-29 | 30-39 | 40-49 | 50-59 | 60-65 | >65 | p-value |
|---|---|---|---|---|---|---|---|---|
| | n=56 | n=155 | n=54 | n=114 | n=237 | n=195 | n=567 | |
| Age | 18.1 (1.6) | 22.7 (1.6) | 36.4 (2.7) | 45.3 (2.8) | 54.5 (2.7) | 62.4 (1.7) | 76.8 (7.9) | <0.001 |
| Weight | 56.5 (11.7) | 58.0 (12.0) | 56.4 (11.0) | 54.9 (8.2) | 56.0 (8.6) | 53.3 (8.2) | 50.8 (9.1) | <0.001 |
| Height | 164.0 (9.9) | 164.1 (8.4) | 159.9 (7.6) | 156.7 (7.8) | 155.6 (7.2) | 152.5 (7.4) | 151.1 (8.4) | <0.001 |
| BMI | 20.91 (3.20) | 21.38 (3.27) | 21.94 (3.18) | 22.37 (2.89) | 23.13 (3.08) | 22.91 (3.24) | 22.24 (3.64) | <0.001 |
| BMD LS (missing=12) | 0.981 (0.149) | 0.973 (0.113) | 0.971 (0.145) | 0.960 (0.132) | 0.865 (0.174) | 0.771 (0.166) | 0.742 (0.193) | <0.001 |
| BMD TH (missing=25) | 1.024 (0.215) | 1.011 (0.142) | 0.992 (0.146) | 1.026 (0.135) | 0.936 (0.145) | 0.862 (0.130) | 0.777 (0.169) | <0.001 |
| BMD FN (missing=22) | 0.977 (0.179) | 0.921 (0.175) | 0.883 (0.118) | 0.892 (0.141) | 0.790 (0.130) | 0.725 (0.118) | 0.662 (0.152) | <0.001 |

Sex-stratified analyses revealed consistent differences between men and women (S1 Table and S2 Table). Men exhibited higher BMD values than women across all sites and age groups. In the 20–29 age group, male FN BMD averaged 0.957 g/cm² compared with 0.871 g/cm² in females. In the oldest age group, male LS BMD was 0.888 g/cm², whereas female LS BMD was 0.696 g/cm². At the FN site, the disparity was even more pronounced, with values of 0.761 g/cm² in men and 0.630 g/cm² in women.

Fig 1, Fig 2, Fig 3, Fig 4, Fig 5, and Fig 6 illustrate the relationship between age and BMD at the LS, TH, and FN sites. In men, BMD curves displayed a plateau from approximately 20–50 years, followed by a gradual decline. In women, BMD decreased sharply after age 50, with the steepest decline observed at the FN.

Peak BMD values and ages of attainment were estimated using age–BMD modeling (Table 2). In women, LS and FN reached maximal values at approximately 30 years of age, while TH peaked later, around 34 years. In men, peak BMD occurred earlier, with LS and FN attaining maximal values during the 20–29 age group and TH peaking at approximately 18 years.

Correlation analyses were conducted to examine the relationships between bone mineral density (BMD) at different skeletal sites and key anthropometric variables, stratified by sex (S3 Table). Across both sexes, age demonstrated strong negative correlations with BMD. In men, the correlation between age and BMD was most pronounced at the femoral neck (r=−0.466), followed by the total hip (r=−0.399) and lumbar spine (r=−0.222). In women, the strength of these associations was even greater, with the femoral neck showing the strongest correlation (r=−0.612), followed by the lumbar spine (r=−0.556) and total hip (r=−0.555).

Positive correlations were observed between body weight and BMD at all sites. In men, weight correlated most strongly with BMD at the total hip (r=0.502) and femoral neck (r=0.473), while in women, the strongest associations were at the total hip (r=0.409) and femoral neck (r=0.363). Similarly, height was positively correlated with BMD, though the strength of association varied by site and sex. In men, height correlated most strongly with the femoral neck (r=0.404), while in women, the strongest correlation was observed at the lumbar spine (r=0.463) sex (S3 Table). Inter-site correlations between BMD values were consistently strong. In men, the correlation between total hip and femoral neck BMD was r=0.822, while in women, the same correlation was even stronger (r=0.883). Correlations between lumbar spine and hip (r=0.675 in men; r=0.737 in women) and between lumbar spine and femoral neck (r=0.583 in men; r=0.736 in women) were also significant sex (S3 Table).

Prevalence estimates of osteoporosis are summarized in Table 3. Crude prevalence was highest at the LS (40.68%), followed by the TH (20.10%) and FN (4.70%). After adjustment, prevalence rates decreased but remained substantial, particularly at the LS (33.58%). Sex-stratified analyses revealed markedly higher prevalence in women compared with

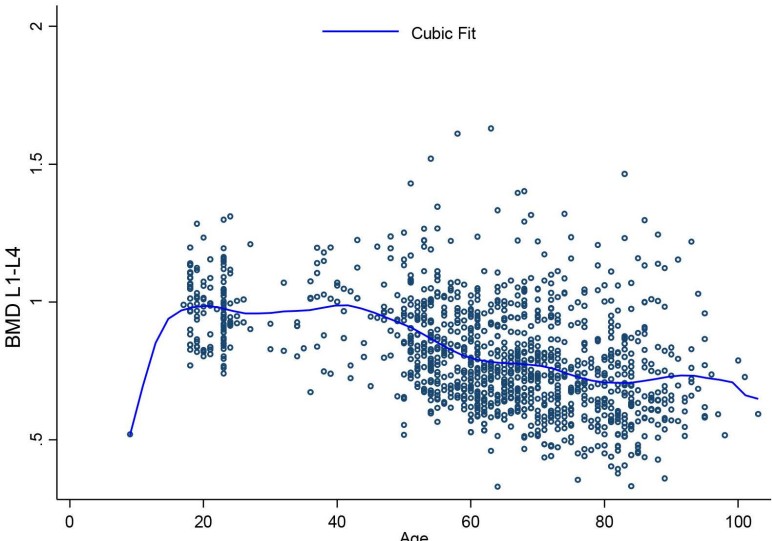

**Fig 1. Relationship between age and lumbar spine BMD.**

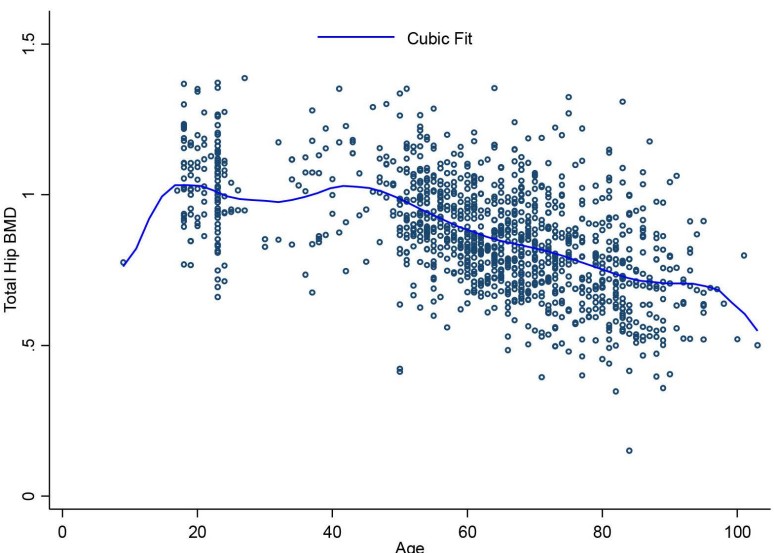

**Fig 2. Relationship between age and total hip BMD.**

men across all sites. At the LS, crude prevalence was 49.22% in women versus 9.72% in men; at the TH, 24.31% versus 5.14%; and at the FN, 5.37% versus 2.32%. Adjusted prevalence rates showed similar disparities, with women consistently exhibiting higher values. Age-stratified data further demonstrated that osteoporosis prevalence increased sharply in women after age 50, reaching 71.43% at the LS, 57.69% at the TH, and 15.82% at the FN in those aged ≥80 years. In men of the same age group, prevalence was considerably lower (14.81% at LS, 11.32% at TH, and 5.56% at FN).

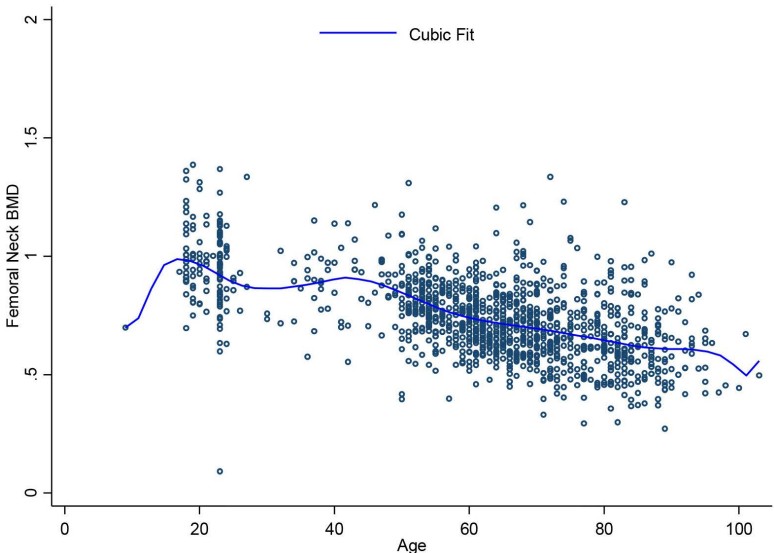

**Fig 3. Relationship between age and femoral neck BMD.**

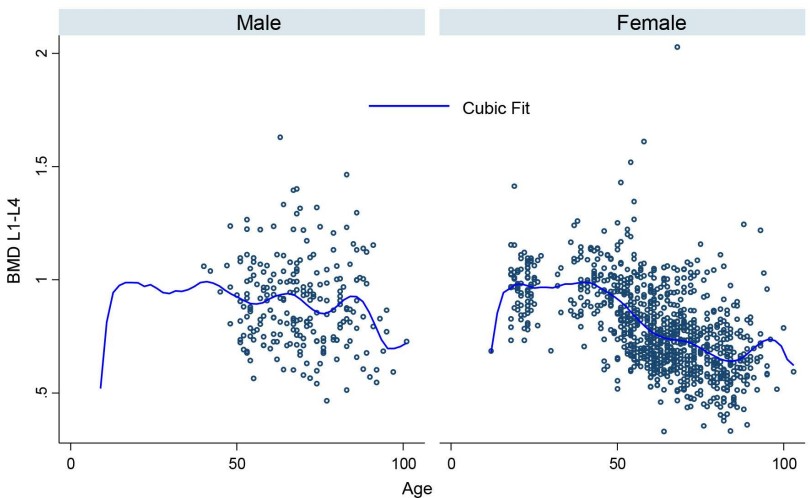

**Fig 4. Relationship between age and lumbar spine BMD in men and women.**

## Discussion

Our study confirms that age, sex, and anthropometric indices are key predictors of BMD, with a marked decline observed in postmenopausal women. Osteoporosis prevalence was highest at the lumbar spine and lowest at the femoral neck. Women exhibited significantly higher osteoporosis rates than men, with prevalence rising sharply with age, especially in those over 60 years.

The findings revealed distinct trajectories, including earlier attainment of peak bone density (PBD) in men than in women, a marked decline in BMD after age 50, and substantial differences in osteoporosis prevalence by site and sex. Notably, the age-BMD modeling in our study indicated that men reached PBD between 20 and 29 years of age, whereas

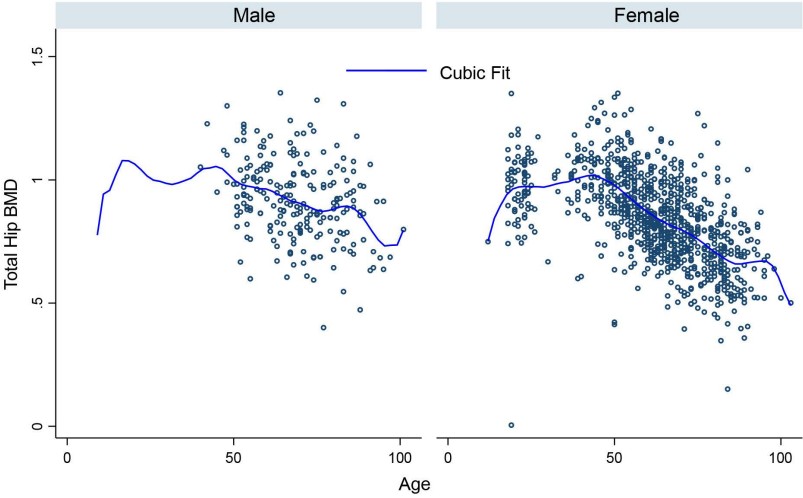

**Fig 5. Relationship between age and total hip BMD in men and women.**

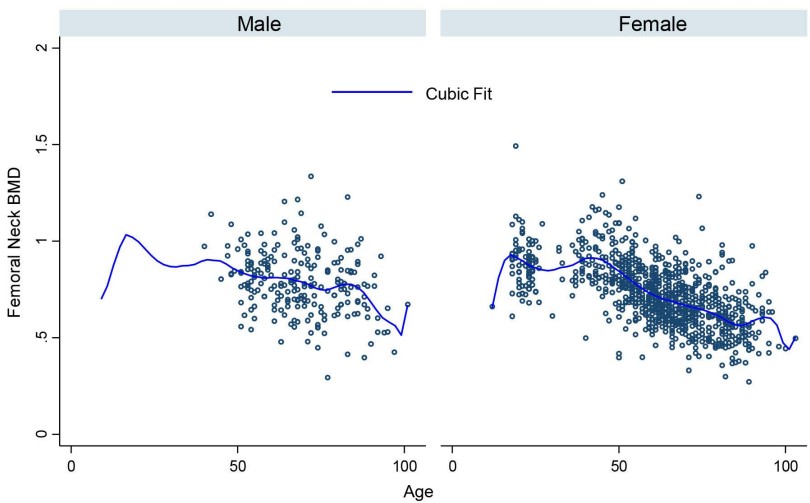

**Fig 6. Relationship between age and femoral neck BMD in men and women.**

**Table 2. Estimate of peak BMD, standard deviation and age of peak BMD.**

| Sex | Female (peak) | | | Male (peak) | | |
|---|---|---|---|---|---|---|
| BMD site | Mean Ref BMD | SD Ref BMD | Age,95%CI or Range | Mean Ref BMD | SD Ref BMD | Age, 95%CI or Range |
| Lumbar Spine (LS) | 0.967 | 0.099 | 29.98(27.29-32.68) | 0.976 | 0.121 | 20-29 years |
| Total Hip (TH) | 0.972 | 0.109 | 34.34(31.63-37.04) | 1.037 | 0.156 | 18.36 (9.31- 27.42) |
| Femoral Neck (FN) | 0.724 | 0.103 | 29.96(26.25-33.68) | 0.957 | 0.203 | 20-29 years |

**Table 3. Crude and Adjusted Prevalence (%) of osteoporosis by site of BMD measurement.**

| BMD site | Femoral Neck (FN), Percent, 95%CI N=978 (F: 763; M: 215) | Total Hip (TH) Percent, 95%CI N=975 (F: 761; M: 214) | Lumbar Spine (LS) Percent, 95%CI N=988 (F: 772; M: 216) |
|---|---|---|---|
| **All (crude)** | **4.70 (3.46-6.22)** | **20.10 (17.63-22.76)** | **40.58 (37.51-43.72)** |
| Male | 2.32 (0.75-5.34) | 5.14 (2.59-9.01) | 9.72 (6.12-14.48) |
| Female | 5.37 (3.88-7.21) | 24.31 (21.30-27.52) | 49.22 (45.64-52.81) |
| **All (Adjusted)** | **2.69 (1.76-3.62)** | **12.77 (10.86-14.69)** | **33.58 (30.43-36.72)** |
| Male | 0.99 (0.08-1.91) | 3.14 (0.82-5.46) | 7.06 (3.54-10.58) |
| Female | 3.19 (2.03-4.35) | 15.52 (13.22-17.83) | 40.77 (37.06-44.49) |
| **Age group (crude)** | | | |
| **Male** | | | |
| 50-59 (%) | 0 | 1.92 | 3.85 |
| 60-69 (%) | 0 | 1.69 | 6.67 |
| 70-79 (%) | 4.00 | 6.00 | 14.00 |
| 80+ (%) | 5.56 | 11.32 | 14.81 |
| **Female** | | | |
| 50-59 (%) | 1.64 | 4.92 | 26.23 |
| 60-69 (%) | 1.17 | 14.06 | 47.10 |
| 70-79 (%) | 6.02 | 30.12 | 56.21 |
| 80+ (%) | 15.82 | 57.69 | 71.43 |

women reached theirs later, at approximately 30 years. When compared with the study by Nguyen HT and colleagues, which reported peak BMD ages in Vietnamese women at approximately 28.5 years for the LS, 29 years for the TH, and 27.7 years for the FN, our results indicate a later peak across all three sites, with PBD occurring at 30 years at the LS, 34 years at the TH, and 30 years at the FN [7,15]. The trend toward later peak BMD attainment (30–34 years) may reflect changes in nutrition, body height, or generational differences. Notably, BMD values in our study were lower than those reported in the 2009 cohort, particularly at the lumbar spine and femoral neck. This reduction may be attributable to decreased physical activity, insufficient calcium or vitamin D intake, or urbanization-related lifestyle shifts that limit habitual mechanical loading. Consequently, these factors may diminish the potential for maximal bone mass accrual compared with previous generations, even though average height has improved.

Regarding comparisons of BMD between sexes, our study found that men consistently had higher BMD than women across all measurement sites and in nearly all age groups. Consistent with Ho-Pham et al. [16], we observed that men consistently exhibited higher BMD than women across all sites. However, a discrepancy exists regarding the timing of peak bone mass; while Ho-Pham et al. reported earlier PBD attainment in women, our data indicate that men reached PBD earlier (20–29 years) compared to women (approx. 30 years) [16]. This difference may reflect a genuine shift in bone development patterns among the current generation of Vietnamese women. Recent anthropometric data indicate that height and growth trajectories in females have changed significantly compared with previous generations, with prolonged periods of linear growth and slower skeletal maturation, potentially shifting the attainment of PBD to older ages. Additionally, urbanized lifestyles characterized by low physical activity, long study or work hours, and extensive sun avoidance, resulting in persistently high rates of vitamin D deficiency among young women, may contribute to delayed bone mineralization and postponement of peak BMD attainment [17]. This trend is consistent with global biological patterns and contributes to updating reference values for the Vietnamese population.

Similar to our findings, sex-specific differences in the age of PBD attainment have been reported in large-scale cohorts from Korea, the United States, and the United Kingdom [18,19]. These variations across populations underscore the complex

interplay of hormonal profiles, genetic background, and lifestyle factors in determining bone development trajectories [20]. We also observed a strong negative association between age and BMD. These findings are comparable with those reported by Nguyen, whose study demonstrated strong inter-site correlations with r values ranging from 0.75 to 0.94 [7].

While the lower prevalence of osteoporosis at the femoral neck compared to Nguyen et al. [7] may reflect regional life-style variations, the persistently high prevalence at the lumbar spine underscores a critical need for targeted screening to prevent vertebral fractures. Prevalence rates in our cohort differ from those reported in regional and international studies [11,21] likely due to variations in sampling methods and reference standards. Clinically, the attainment of PBD by age 30 highlights a pivotal window for lifestyle interventions to maximize bone accretion in young adults. Furthermore, the steep decline in BMD post-menopause supports the prioritization of early screening for women aged 50 and older, particularly at the lumbar spine.

Several limitations merit consideration. First, the cross-sectional design precludes causal inferences regarding bone loss trajectories. Second, despite rigorous exclusion criteria, residual confounding from unmeasured lifestyle factors such as specific dietary calcium intake and physical activity intensity cannot be ruled out. Finally, as a single-center study, the gener-alizability of these findings to the broader Vietnamese population requires further validation through multi-center initiatives.

Future research should prioritize prospective cohort designs to track longitudinal bone changes and integrate compre-hensive lifestyle data to refine predictive models. Building on these models, future initiatives should explore rapid, highly specific detection strategies to develop a 'real-time detection and AI interpretation' system for osteoporosis risk, which could significantly shorten the screening cycle [22, 23]. Expanding data collection across diverse geographic regions will be essential to establish a truly representative national reference standard.

## Conclusion

This study provides updated estimates of peak bone density and the predicted age at which peak BMD is achieved in Vietnamese men and women. The findings demonstrate apparent differences in PBD by age, sex, and measurement site, with men reaching PBD earlier than women, and women experiencing a more rapid decline in BMD following menopause. BMD at the lumbar spine and femoral neck showed strong associations with age and anthropometric factors, reflecting the physiological progression of bone loss and fracture risk. These results contribute essential baseline data to support standardized diagnostic criteria and risk assessment for osteoporosis. The evidence also underscores the importance of optimizing PBD during young adulthood and implementing early screening in high-risk groups. Overall, this study provides a new reference dataset for the Vietnamese population and lays the groundwork for the application of predictive modeling and longitudinal research to strengthen future osteoporosis prevention and intervention strategies.

## Supporting information

**S1 Table. Characteristics of study subjects (Male, n = 410).**
(DOCX)

**S2 Table. Characteristics of study subjects (Female, n = 968).**
(DOCX)

**S3 Table. Correlation analysis of varibles stratified by gender.**
(DOCX)

## Author contributions

**Conceptualization:** Hoang Thanh Van Nguyen, Thi Thu Ha Tran, Huu Quyet Le, Hoang Minh Nguyen, Thi Hong Van Le.

**Data curation:** Hoang Thanh Van Nguyen, Thi Thu Ha Tran, Huu Quyet Le, Thi Hong Van Le.

 

**Formal analysis:** Hoang Thanh Van Nguyen, Thi Thu Ha Tran, Huu Quyet Le, Hoang Minh Nguyen, Thi Hong Van Le.

**Funding acquisition:** Hoang Thanh Van Nguyen, Hoang Minh Nguyen.

**Investigation:** Hoang Thanh Van Nguyen.

**Methodology:** Hoang Thanh Van Nguyen, Huu Quyet Le, Hoang Minh Nguyen.

**Project administration:** Hoang Thanh Van Nguyen.

**Resources:** Hoang Thanh Van Nguyen, Hoang Minh Nguyen, Thi Hong Van Le.

**Software:** Hoang Thanh Van Nguyen, Hoang Minh Nguyen, Thi Hong Van Le.

**Supervision:** Hoang Thanh Van Nguyen.

**Validation:** Hoang Thanh Van Nguyen, Huu Quyet Le.

**Visualization:** Hoang Thanh Van Nguyen, Thi Thu Ha Tran, Hoang Minh Nguyen.

**Writing – original draft:** Hoang Thanh Van Nguyen, Thi Thu Ha Tran, Hoang Minh Nguyen, Thi Hong Van Le.

**Writing – review & editing:** Hoang Thanh Van Nguyen.

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
