## [Decision Letter · Decision Letter 0]

18 Feb 2026

PONE-D-25-65964Assessment of Peak Bone Mineral Density and Its Associated Factors in Vietnamese Adults: A Cross-Sectional StudyPLOS One

Dear Dr. Nguyen,

Thank you for submitting your manuscript to PLOS ONE. After careful consideration, we feel that it has merit but does not fully meet PLOS ONE’s publication criteria as it currently stands. Therefore, we invite you to submit a revised version of the manuscript that addresses the points raised during the review process.

We look forward to receiving your revised manuscript.

Kind regards,

Melissa Orlandin Premaor, M.D., Ph.D

Academic Editor

PLOS One

2. We notice that your supplementary tables are included in the manuscript file. Please remove them and upload them with the file type 'Supporting Information'. Please ensure that each Supporting Information file has a legend listed in the manuscript after the references list.

Reviewers' comments:

Reviewer's Responses to Questions

**Comments to the Author**

1. Is the manuscript technically sound, and do the data support the conclusions?

Reviewer #1: Yes

2. Has the statistical analysis been performed appropriately and rigorously? 

Reviewer #1: Yes

3. Have the authors made all data underlying the findings in their manuscript fully available?

The PLOS Data policy requires authors to make all data underlying the findings described in their manuscript fully available without restriction, with rare exception (please refer to the Data Availability Statement in the manuscript PDF file). The data should be provided as part of the manuscript or its supporting information, or deposited to a public repository. For example, in addition to summary statistics, the data points behind means, medians and variance measures should be available. If there are restrictions on publicly sharing data—e.g. participant privacy or use of data from a third party—those must be specified.requires authors to make all data underlying the findings described in their manuscript fully available without restriction, with rare exception (please refer to the Data Availability Statement in the manuscript PDF file). The data should be provided as part of the manuscript or its supporting information, or deposited to a public repository. For example, in addition to summary statistics, the data points behind means, medians and variance measures should be available. If there are restrictions on publicly sharing data—e.g. participant privacy or use of data from a third party—those must be specified.requires authors to make all data underlying the findings described in their manuscript fully available without restriction, with rare exception (please refer to the Data Availability Statement in the manuscript PDF file). The data should be provided as part of the manuscript or its supporting information, or deposited to a public repository. For example, in addition to summary statistics, the data points behind means, medians and variance measures should be available. If there are restrictions on publicly sharing data—e.g. participant privacy or use of data from a third party—those must be specified.requires authors to make all data underlying the findings described in their manuscript fully available without restriction, with rare exception (please refer to the Data Availability Statement in the manuscript PDF file). The data should be provided as part of the manuscript or its supporting information, or deposited to a public repository. For example, in addition to summary statistics, the data points behind means, medians and variance measures should be available. If there are restrictions on publicly sharing data—e.g. participant privacy or use of data from a third party—those must be specified.

Reviewer #1: Yes

4. Is the manuscript presented in an intelligible fashion and written in standard English?

Reviewer #1: Yes

5. Review Comments to the Author

Reviewer #1: This study systematically established cross-sectional data on the peak bone density (PBD) of adult Vietnamese individuals and the influencing factors, filling the gap in population-specific reference data and having significant value for the prevention and control of osteoporosis. The methods were standardized and the statistics were reasonable. The results revealed key findings such as higher PBD in men than in women and a higher prevalence of lumbar spine disorders. However, there were limitations:

1.The cross-sectional design makes it difficult to establish causality;

2. The generalizability of the single-center sample is limited;

3.Uncontrolled confounding factors such as dietary calcium intake and physical activity were not controlled. It is recommended to conduct multi-center prospective studies for verification to enhance the national representativeness; references can be made to (Nati. Sci. Rev., 2022, 9, nwac104., ACS nano, 2023, 17(13): 12903-12914.) for strategies in non-amplification, rapid, and highly specific detection, and it is recommended to develop a "real-time detection + AI interpretation" system for osteoporosis risk, shortening the screening cycle.

6. PLOS authors have the option to publish the peer review history of their article (what does this mean?). If published, this will include your full peer review and any attached files.). If published, this will include your full peer review and any attached files.). If published, this will include your full peer review and any attached files.). If published, this will include your full peer review and any attached files.

...

Reviewer #1: No

---

## [Author Response · Author response to Decision Letter 1]

2 Mar 2026

Reply 1. We have gone through the guidelines and revised it thoughtly.

2. We notice that your supplementary tables are included in the manuscript file. Please remove them and upload them with the file type 'Supporting Information'. Please ensure that each Supporting Information file has a legend listed in the manuscript after the references list.

Reply 2. We have moved and uploaded as 'Supporting Information”

Reviewer #1: This study systematically established cross-sectional data on the peak bone density (PBD) of adult Vietnamese individuals and the influencing factors, filling the gap in population-specific reference data and having significant value for the prevention and control of osteoporosis. The methods were standardized and the statistics were reasonable. The results revealed key findings such as higher PBD in men than in women and a higher prevalence of lumbar spine disorders.

However, there were limitations:

1.The cross-sectional design makes it difficult to establish causality;

2. The generalizability of the single-center sample is limited;

3.Uncontrolled confounding factors such as dietary calcium intake and physical activity were not controlled. It is recommended to conduct multi-center prospective studies for verification to enhance the national representativeness; references can be made to (Nati. Sci. Rev., 2022, 9, nwac104., ACS nano, 2023, 17(13): 12903-12914.) for strategies in non-amplification, rapid, and highly specific detection, and it is recommended to develop a "real-time detection + AI interpretation" system for osteoporosis risk, shortening the screening cycle.

Reply 3: We sincerely thank you for your thoughtful comments and insightful analysis of our manuscript. All the points raised are reasonable, and we have acknowledged them in the limitations section of the discussion. We have also incorporated your suggestions regarding future research directions (page 13, last paragraph).

“Future research should prioritize prospective cohort designs to track longitudinal bone changes and integrate comprehensive lifestyle data to refine predictive models. Building on these models, future initiatives should explore rapid, highly specific detection strategies to develop a 'real-time detection and AI interpretation' system for osteoporosis risk, which could significantly shorten the screening cycle [22,23]. Expanding data collection across diverse geographic regions will be essential to establish a truly representative national reference standard”

---

## [Editor Report · Decision Letter 1]

16 Mar 2026

Assessment of Peak Bone Mineral Density and Its Associated Factors in Vietnamese Adults: A Cross-Sectional Study

PONE-D-25-65964R1

Dear Dr. Nguyen,

We’re pleased to inform you that your manuscript has been judged scientifically suitable for publication and will be formally accepted for publication once it meets all outstanding technical requirements.

Kind regards,

Melissa Orlandin Premaor, M.D., Ph.D

Academic Editor

PLOS One
---

## [Editor Report · Acceptance letter]

PONE-D-25-65964R1

PLOS One

Dear Dr. Nguyen,

I'm pleased to inform you that your manuscript has been deemed suitable for publication in PLOS One. Congratulations! Your manuscript is now being handed over to our production team.

Kind regards,

on behalf of

Dr. Melissa Orlandin Premaor

Academic Editor

PLOS One